# Understanding no-show behaviour for cervical cancer screening appointments among hard-to-reach women in Bogotá, Colombia: A mixed-methods approach

David Barrera Ferro[1,2]*, Steffen Bayer[1], Laura Bocanegra[3], Sally Brailsford[1], Adriana Díaz[2‡], Elena Valentina Gutiérrez-Gutiérrez[4‡], Honora Smith[5]

1 Southampton Business School, University of Southampton, Southampton, United Kingdom,
2 Departamento de Ingeniería Industrial, Pontificia Universidad Javeriana, Bogotá, Colombia, 3 Secretaría Distrital de Salud, Bogotá, Colombia, 4 School of Industrial Engineering, Universidad del Valle, Cali, Colombia, 5 Mathematical Sciences, University of Southampton, Southampton, United Kingdom

☺ These authors contributed equally to this work.
‡ AD and EVGG also contributed equally to this work.
* barrera-o@javeriana.edu.co

**Data Availability Statement:** The datasets generated and analysed during the current study

## Abstract

The global burden of cervical cancer remains a concern and higher early mortality rates are associated with poverty and limited health education. However, screening programs continue to face implementation challenges, especially in developing country contexts. In this study, we use a mixed-methods approach to understand the reasons for no-show behaviour for cervical cancer screening appointments among hard-to-reach low-income women in Bogotá, Colombia. In the quantitative phase, individual attendance probabilities are predicted using administrative records from an outreach program (N = 23384) using both LASSO regression and Random Forest methods. In the qualitative phase, semi-structured interviews are analysed to understand patient perspectives (N = 60). Both inductive and deductive coding are used to identify first-order categories and content analysis is facilitated using the Framework method. Quantitative analysis shows that younger patients and those living in zones of poverty are more likely to miss their appointments. Likewise, appointments scheduled on Saturdays, during the school vacation periods or with lead times longer than 10 days have higher no-show risk. Qualitative data shows that patients find it hard to navigate the service delivery process, face barriers accessing the health system and hold negative beliefs about cervical cytology.

## 1. Introduction

Despite being highly preventable, cervical cancer is the fourth leading cause of cancer death in women: in 2020, 341,831 women died worldwide of this disease [1]. Additionally, incidence and early mortality rates of this type of cancer are associated with limited education and poverty [2–5]. While in North America, age standardised rates (ASR) of incidence and mortality

are owned and managed by SDS. Therefore, we are not able to share any raw data without violating the terms of the ethics approvals. Further, following a recommendation from SDS Ethics Committee for Health Research, data are not publicly available as they include sensitive and potentially identifying patient information. However, they are available from SDS (subject to further ethical approval) on reasonable request. Please contact Leidy Castaneda (ljcastaneda@saludcapital.gov.co).

**Funding:** DBF research is funded by a PhD scholarship (Grant 3929446) from the healthcare research stream of the program Colombia Científica – Pasaporte a la Ciencia, granted by the Colombian Institute for Educational Technical Studies Abroad (Instituto Colombiano de Crédito Educativo y Estudios Técnicos en el Exterior, ICETEX). There was no additional external funding received for this study. The funder had no role in study design, data collection and analysis, decision to publish, or preparation of the manuscript.

**Competing interests:** The authors have declared that no competing interests exist.

are 6.1 and 2.1 per 100,000 women respectively [6], in Colombia these indicators are 14.9 and 7.4 per 100,000 women [7]. Therefore, early diagnosis and health education have been identified as key components in the effort to advance cervical cancer control worldwide [8]. However, in many lower and middle-income countries (LMICs), screening programs still face implementation challenges [9, 10]. In Colombia, this disease is the leading reason of death by cancer among women between 30 and 59 years old in the country, and its burden continues to be a concern [11, 12].

In Bogotá, as part of a preventive-care strategy called *Acciones Colectivas en Salud* (ACS), the District Secretariat of Health (*Secretaría Distrital de Salud*, SDS) instituted a program to increase cervical cancer cytology uptake among hard-to-reach low-income women. Under this program, a group of community workers visit women who have not taken a cytology test during the last year, conduct basic training in cervical cancer risks and schedule a cytology appointment for them at the nearest healthcare facility. Over the last two years, the program has increased its coverage; however, no-show rates have reached levels of 46%. Therefore, no-show behaviour represents a challenge for program managers from both effectiveness and efficiency perspectives [13, 14]. In this context, more information is needed to support the design of population-based strategies.

Quantitative and qualitative approaches have been used in recent studies to understand cancer screening uptake rates in developing countries. Black et al. [2] and Nuche-Berenguer and Sakellariou [15] review quantitative studies conducted in Uganda and Latin America, respectively. Both studies conclude that more research is needed in order to understand lower participation of low-income population in screening programs. To the best of our knowledge, no review of qualitative approaches has been published at the time of writing. However, qualitative studies have been undertaken in Tanzania [16], Ethiopia [17], Botswana [18] and Nigeria [19], among others. In these four studies, detailed conversations with patients have enabled context-dependent barriers to be identified. Further, researchers conclude that interventions to increase cervical cancer screening uptake should be tailored to the local population, taking into account aspects such as levels of health education, religious affiliations, and personal beliefs of the patients. Although the emphasis on evidence-based research might explain the dominance of quantitative methods, the contribution of qualitative methods in health research is now increasingly accepted [20].

In this context, mixed-methods research has the potential to provide more complete information regarding no-show behaviour [10, 16]. According to Wisdom [21], the combined use of quantitative and qualitative methods can provide a more comprehensive picture of health services by capitalizing on the strengths of both approaches. Despite being a relatively new area, Guetterman et al. [22] found that there is an increasing awareness of the relevance of mixed-methods research in order to address population and behavioural health problems. French et al. [23], for example, used regression models to identify characteristics of children who missed their appointments and conducted phone interviews with GPs in order to understand their role and perceptions regarding low attendance levels.

The aim of this study is to understand this no-show behaviour by combining prediction and interpretation approaches. The prediction approach is premised on the idea that it is possible to use routinely collected historical data to produce a numerical estimate of the attendance probability for each individual patient. However, the retrospective nature and limitations of such data make it impossible to identify the reasons that could lead to a missed appointment [24, 25]. The aim of the interpretation approach is to understand the phenomenon by studying patients' perceptions and their decision-making processes [26]. Therefore, we use a qualitative approach to undertake an in-depth exploration of the perceived barriers to attendance [27].

## 2. Methods

In this section, we first present the study context. Next, we discuss how the quantitative and qualitative phases interact and inform our conclusions. Then, for each phase, we describe the process of data collection and the analytical approach adopted. When pertinent, RECORD (The REporting of studies Conducted using Observational Routinely-collected health Data) [28] and SRQR (Standards for Reporting Qualitative Research) [29] guidelines are followed. Pontificia Universidad Javeriana (Faculty of Engineering's Research and Ethics Committee: FID-19-107), SDS (Ethics Committee for Health Research 2019EE47807) and the University of Southampton (Faculty of Social Sciences' Ethics and Research Committee ERGO ID 48583. A1) granted ethical approval for this study.

### 2.1. Study context

In Colombia, the cervical cancer screening program covers women between 25 and 65 years old, or younger in the presence of some risk factors [30]. Currently, this program primarily relies on Pap smear tests following a 1-1-3 scheme [31, 32]. This means that women should undergo annual cytology tests, and then change to a three-year interval after two consecutive negative results. Additionally, the screening is included in the national health insurance scheme and hence no out-of-pocket payment is required. Recent legislation has adopted the Human Papilloma Virus (HPV) test for women between 30 and 65 years old, as screening strategy [31]. However, at the time of writing, we were not able to find any consolidated report about the HPV test piloting in the country.

In Bogotá, the cervical cancer screening component of ACS is designed to cover hard-to-reach women. For this program, SDS considers a woman to be hard-to-reach if despite being eligible, she has not undergone a Pap smear test over the last year. Additionally, to prioritize resource allocation for social programs, SDS uses a nation-wide adopted scoring system that classifies low-income citizens into four categories. The SISBEN [Identification System of Potential Beneficiaries of Social Programs *(Sistema de Identificación de Potenciales Beneficiarios de Programas Sociales)*] score ranges from 0 (extreme poverty) to 100 (wealthy) and is computed using self-reported information related to health, education, and housing, among others [33]. ACS covers approximately 18% of the population with the lowest SISBEN score [34].

### 2.2. Integration approach

According to Fetters et al. [35], in mixed-methods health research, integration might occur at three different levels: design, methods and interpretation. From a research design perspective, we use qualitative data to understand specific aspects of the quantitative findings. This is called an explanatory sequential approach. At the methods level, the quantitative findings inform the sample definition for the qualitative component. Therefore, at the methods level, we seek an integration through building. Lastly, results from both phases are reported independently and we analyse aspects of the problem that can be better understood as a result of integration. This is called integration through narrative using a continuous approach.

In our case, quantitative data are used to predict individual attendance probabilities and qualitative data are used to understand the patient experience. In order to build prediction models, we conduct statistical analysis using administrative records. Then, a series of semi-structured interviews are performed to understand the patient perspective regarding no-show behaviour. Therefore, in this research, integration occurs at two points: i) a patient is invited to take part of the interviews only if her no-show risk, according to the prediction models, was medium or high and ii) the results of the interviews are used to enhance the analysis of the prediction models.

## 2.3. Predicting attendance probabilities: The quantitative phase

We analysed data collected routinely by program managers (in SDS) to assess the performance of ACS. Between January 2017 and December 2019, appointments were scheduled for 23384 women aged between 21 and 65 years old. In each case, the outcome–show or no-show–was recorded. Table 1 presents the list of variables, grouped into two categories: patient and appointment-related information. We did not have access to poverty level data, marital status, or number of children for individual patients: these data are not held by SDS. For age and lead time we used decision trees to build categorical variables maximizing information value. This means that the categories (the number and the limits) were automatically selected by the algorithm to maximize inter-category difference and minimize intra-categories difference. This approach has also been found to generate more stable models [36]. SDS granted access to a fully anonymized database for our analysis. The data were accessed in August 2019 (all records from January 2017 to July 2019) and February 2020 (all records from August to December 2019). From this database, we randomly generated training (70%) and test (30%) sets.

To estimate the probability of attendance, two well-known models were implemented, Least Absolute Shrinkage and Selection Operator (LASSO) regression [37] and Random Forests (RF) [38]. Recent applications of LASSO in healthcare research include prediction of mortality rates [39] and medication adherence [40], among others. Additionally, for classification problems, RFs are less sensitive to outliers and eliminate the risk of overfitting [41] and thus improve the accuracy of the model. We conducted a parametric analysis on the penalization constant of the LASSO model and selected the one that maximizes the Area Under the Receiver Operating Curve (AUROC) while minimizing the number of selected variables. For classification proposes, a value of one was assigned to those patients attending their appointments. Therefore, higher odds ratios mean higher attendance probabilities.

To validate the model, we randomly divided the training set into 10 groups, used nine groups for training, and the other for testing. Then, the testing group was iteratively changed, and the procedure was repeated ten times, resulting in 100 experiments. This is called a 10-by-10 cross validation process (10-by-10 CV). We used the LASSO results to select the features included in the RF, optimized parameters using 30% of the training set and performed a 10-by-10 CV. The performance of both models was assessed using the average and standard deviation of the AUROC score over the 100 experiments. LASSO and RF Scikit-Learn's implementations were used for our analysis [42].

## 2.4. Understanding patient experience: The qualitative phase

The aim of the qualitative phase is to understand the patient experience and reasons for health-seeking behaviour. Data were collected through semi-structured interviews using purposeful sampling [43, 44]. We focused our analysis on patients with higher no-show risk, as their views can provide relevant information to design behavioural interventions [45].

**Table 1. Variables used for prediction models.**

| Category | Variable | Description |
|---|---|---|
| Patient | Age | Age of the patient at the moment of the appointment (years) |
| | Zone | Area of the city where the patient lives |
| | Poverty | Percentage of population living in poverty within the patient zone |
| Appointment | Lead time | Elapsed time between the date of the home visit and the appointment date (days) |
| | Month | Month in which the appointment was scheduled |
| | Day | Day of the week in which the appointment was scheduled |

Therefore, patients who met the following three eligibility criteria were considered: i) having received a home visit and an appointment scheduled between October and December of 2019 (3140 patients), ii) additionally, had been classified as a medium or high no-show risk according to the prediction models (1099 patients) and iii) additionally, had failed to keep their appointments (857 patients). Program managers provided a list of 100 randomly selected patients that met the criteria; we were able to reach 75 patients by phone and, of these, 15 declined to participate.

Five community workers collected data using phone interviews in Spanish, between January and February 2020. A nine-item interview guide was designed using relevant literature and discussed with public health specialists and community workers at SDS, in one workshop (see S1 Appendix). Before starting data collection, training took place in two workshops where the research project was presented, and each item of the interview guide was discussed. Since these community workers perform home visits as part of their normal jobs, they have had previous training on working with vulnerable populations and discussing health-related topics. In each phone call, basic information of the project was provided, the patient was invited to take part of the study and oral informed consent was obtained. Patients authorized the conversations to be recorded. A research assistant performed verbatim transcriptions of the audio files and one of the researchers checked quality of the transcription.

Data analysis was conducted in Spanish and facilitated using the Framework method [46]. This is a well-established method for health multidisciplinary research projects, as it enables large data sets to be organized and compared [47]. It has been argued that this method is particularly appropriate for research questions in which different views, in relation to a topic, are analysed and therefore a descriptive overview is required [48]. Table 2 provides basic information of our approach in each of the seven stages proposed by Gale et al. [48] to analyse qualitative healthcare data using the Framework method.

To design the analytical framework, both inductive and deductive analysis were used. On the one hand, inductive coding enabled the identification of under-researched topics, as categories emerged from the data [49]. On the other hand, deductive coding facilitated to take advantage of findings that have been previously documented in the research topic by using categories derived from the literature and prior experience [44]. The coding team (DB, AD, and VG) developed inductive first-order categories (i.e., emerging themes) using 10 interviews. Each researcher produced a preliminary list of categories, and these lists were analysed and discussed until consensus was reached.

**Table 2. Seven stages for analysis using the framework method.**

| | Stage | Our project |
|---|---|---|
| 1 | Transcription | A research assistant performed verbatim transcriptions of audio files and one of the authors checked quality of the transcription. |
| 2 | Familiarisation | Ten interviews were analysed by the coding team composed of three researchers. |
| 3 | Coding | As a pilot study, each member of the coding team analysed the first 20 audios and notes were compared. |
| 4 | Developing a working analytical framework | Both inductive and deductive analysis are performed. |
| 5 | Applying the analytical framework | Two members of the coding team coded each interview (n = 60) using NVivo 12. |
| 6 | Charting data into the framework matrix | Computer-Aided Qualitative Analysis Software (NVivo 12) |
| 7 | Interpreting the data | Several virtual meetings. |

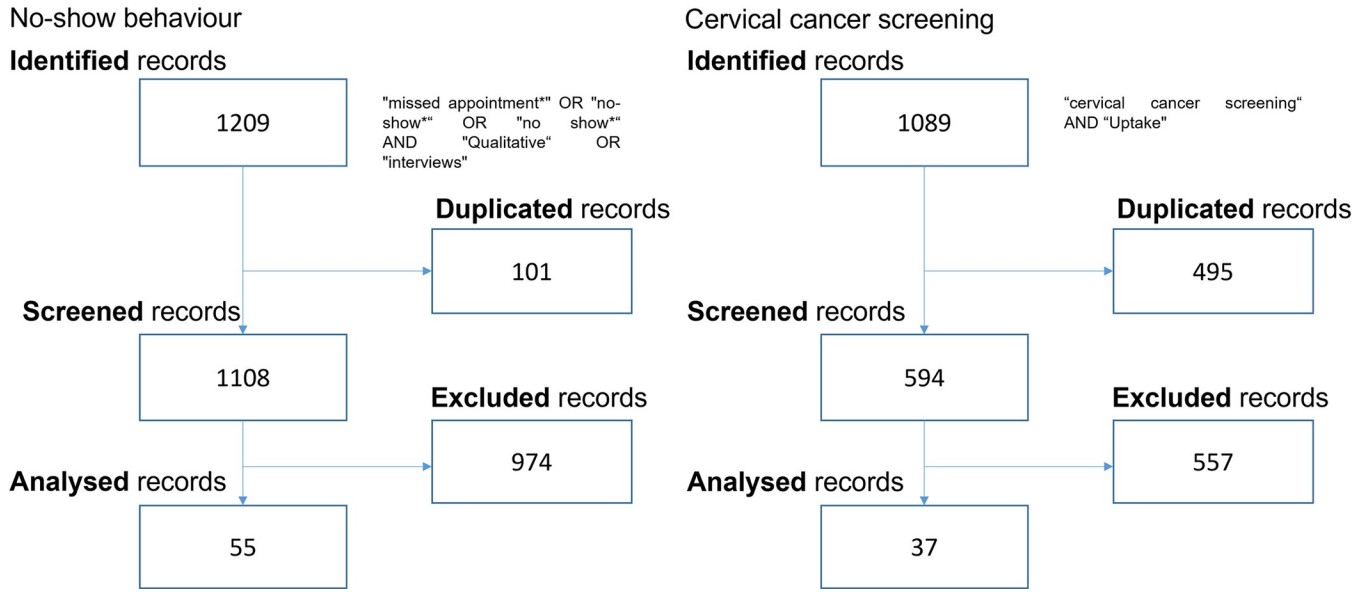

**Fig 1. Literature searches.**

Two literature searches were conducted, using SCOPUS and PubMed databases, to identify deductive first-order categories. Fig 1 provides details of each review using the PRISMA guidelines [50]. We decided to limit our search using the title, abstract and key words option in SCOPUS and the title and abstract option in PubMed. First, we targeted journal papers, published in English, that use qualitative analysis in order to understand no-show behaviour in healthcare. We identified 55 papers published between 2004 and 2021. We note that of these 55, 40 were published after 2015 and only eight are on the topic of no-show behaviour in developing countries. Second, we aimed at identifying qualitative works studying cervical cancer screening uptake. We identified 37 papers published between 2005 and 2021. The majority of these works (62%) were conducted in developing country contexts.

To build second order categories (i.e. groups of first-order categories), we adopt the Health Belief Model (HBM) [51] as a conceptual framework. The use of the HBM to understand behaviours and design population-based interventions in preventive care has been widely documented [52]. The main idea is that the adoption of protective behaviours can be explained by what the patient perceives in terms of severity, benefits, susceptibility, and barriers. Therefore, we group the first-order categories using these constructs. Table 3 presents the resulting 44 categories of the analytical framework. Additionally, a description of each category and the list of references supporting the deductive categories are provided in S2 Appendix. The ten inductive categories were included at this stage. We believe that readers interested in healthcare no-show behaviour could find this framework useful to analyse qualitative data or inform instrument design in other contexts.

The three researchers of the coding team were involved in the analysis of each interview. First, we conducted a pilot using 20 transcriptions. In the pilot, each researcher coded independently and made notes of possible adjustments needed in the framework. These adjustments were then discussed in a joint meeting and a new version of the framework produced. Secondly, for each interview, two researchers were assigned to code independently and generate a preliminary version of the framework matrix using NVivo. Then, the third researcher analysed the resulting categories, identified differences, made notes, and formed a

**Table 3. Analytical framework categories.**

| Second order | First order | | |
|---|---|---|---|
| Barriers | Access | | |
| | | 1 | Financial stress |
| | | 2 | Inconvenient appointment slots |
| | | 3 | Long lead times |
| | | 4 | Geographical access |
| | | 5 | Work Commitments |
| | Service delivery | | |
| | | 6 | Bad experiences with service delivery |
| | | 7 | Bad experiences with home visit |
| | | 8 | Communication |
| | | 9 | Dismissive staff |
| | | 10 | Lack of flexibility in service delivery |
| | | 11 | Lack of information during the home visit |
| | | 12 | Multiple appointments |
| | | 13 | Poor care quality |
| - | | 14 | Prefers to use other care |
| | | 15 | Process design |
| | Personal | | |
| | | 16 | Family care |
| | | 17 | Forgetfulness |
| | | 18 | Health issues |
| | | 19 | Lack of network support |
| | | 20 | Language |
| | | 21 | Migration |
| | | 22 | Other priorities |
| | | 23 | Religion |
| | | 24 | Travel |
| Barriers | Protective behaviour | | |
| | | 25 | Anxiety |
| | | 26 | Non-compliance with requirements |
| | | 27 | Discomfort |
| | | 28 | Embarrassment |
| | | 29 | Gender of the health provider |
| | | 30 | Pain |
| | | 31 | Peer influence |
| Benefits | Protective Behaviour | | |
| | | 32 | Cancer diagnosis |
| | | 33 | Health |
| | | 34 | Lack of perceived benefits |
| | | 35 | Lack of knowledge |
| | | 36 | Screening program |
| | Service delivery | | |
| | | 37 | Satisfaction (home visit) |
| | | 38 | Satisfaction (service delivery) |
| Susceptibility | | 39 | Perceived susceptibility |
| | | 40 | Denial |
| Severity | | 41 | Fear of a bad result |

(*Continued*)

**Table 3.** (Continued)

| Second order | First order | |
|---|---|---|
| | 42 | Fear of side effects |
| | 43 | Only uses emergency care |
| | 44 | Severity of the consequences |

recommendation. All differences were analysed in joint meetings until consensus was reached among the three researchers. We were able to reach thematic saturation with our initial sample of 60 interviews, therefore no second round of interviews was required [53]. Lastly, a final version of the matrix was generated to inform discussions among all researchers.

## 3. Results

This section starts with an analysis of the LASSO regression results and an assessment of the accuracy improvements achieved by RF. The qualitative findings then follow, with a discussion of the categories resulting from content analysis using our analytical framework.

### 3.1. Quantitative results

Table 4 presents the results of the LASSO regression model. This model has a moderate discriminatory power, and its results are not sensitive to the sample. The average AUROC score is 0.65 with a standard deviation of 0.001. This could indicate that the non-linear component of the relationship between the variables and the attendance probability is high. It is also possible that including additional patient information could lead to better performance. Variables such as income and education levels have been found to be good predictors of attendance for cervical cancer screening [54, 55]. However, our aim was to leverage routinely available data to inform patient prioritization by SDS. Therefore, the LASSO results are used to understand the characteristics of patients with higher no-show risk and to select the variables that should be used in the RF model.

There is a relationship between patient-related variables and attendance probability. The odds ratios for the zone in which the patient lives range from 0.47 (zone 65) to 4.48 (zone 11). Additionally, patients living in zones where poverty affects less than 18% of the population are three times more likely to attend their appointments than those living in the remaining zones. Lastly, the younger the patient, the higher her no-show risk.

Table 4 also shows a relationship between appointment-related variables and attendance probability. Regarding the appointment month, school vacation periods (January, March, June, and December) have lower odds ratios. Additionally, while patients are more likely to attend appointments on Sundays, Saturday appointments have a higher no-show risk. This might indicate that the requirement to take time off from work could act as a barrier to cytology uptake. Lastly, we find that longer lead times increase the risk of no-show.

In terms of AUROC score, the use of RF adds value to the classification. Average score of the RF is 0.84 (29% higher that the LASSO score) with a standard deviation of 0.01. However, LASSO results are less sensitive to the sample and hence potentially more reliable when used for different data. One practical implication of an improvement in accuracy relates to the design of interventions to reduce no-show behaviour. Mass interventions aimed at the whole population are generally not cost-effective [56] since a significant proportion of patients are likely to attend with no intervention at all. Initiatives can be made more cost-effective, and hence financially sustainable, by attempting to target those patients at greatest risk of no-show

**Table 4. Results of the LASSO regression model.**

| Variable | | Coefficient | | | Odds Ratio |
|---|---|---|---|---|---|
| | | Average | Percentile 5th | Percentile 95th | Average |
| Age (years) | | | | | |
| | [21, 27] | -0.82 | -0.85 | -0.79 | **0.44** |
| | [27, 45] | -0.44 | -0.46 | -0.42 | 0.64 |
| | > 45 | | | | **1.00** |
| Zone | | | | | |
| | 11. San Cristobal | 1.50 | 1.42 | 1.60 | **4.47** |
| | 55. Diana Turbay | 1.28 | 1.20 | 1.35 | 3.60 |
| | 57. Gran Yomasa | -0.79 | -0.85 | -0.72 | **0.45** |
| | 65. Arborizadora | -0.75 | -0.87 | -0.64 | 0.47 |
| Poverty | | | | | |
| | [0%, 18%] | 1.10 | 1.05 | 1.16 | **3.01** |
| | > 18% | | | | **1.00** |
| Lead time (days) | | | | | |
| | [0, 9.0] | 0.46 | 0.44 | 0.49 | **1.58** |
| | [9.0, 10] | 0.13 | 0.08 | 0.19 | 1.14 |
| | > 10 | | | | **1.00** |
| Day | | | | | |
| | Sunday | 0.96 | 0.86 | 1.09 | **2.61** |
| | Monday | -0.02 | -0.03 | -0.01 | 0.98 |
| | Tuesday | | | | 1.00 |
| | Wednesday | | | | 1.00 |
| | Thursday | 0.06 | 0.03 | 0.08 | 1.06 |
| | Friday | -0.07 | -0.09 | -0.05 | 0.93 |
| | Saturday | -0.20 | -0.23 | -0.17 | **0.82** |
| Month | | | | | |
| | January | -0.19 | -0.22 | -0.15 | 0.83 |
| | February | 0.07 | 0.03 | 0.11 | 1.07 |
| | March | -0.29 | -0.34 | -0.26 | 0.74 |
| | April | 0.38 | 0.32 | 0.45 | **1.46** |
| | May | 0.04 | 0.01 | 0.08 | 1.04 |
| | June | -0.41 | -0.45 | -0.36 | 0.67 |
| | July | 0.07 | 0.03 | 0.10 | 1.07 |
| | August | -0.03 | -0.05 | -0.01 | 0.97 |
| | September | | | | 1.00 |
| | October | -0.14 | -0.17 | -0.11 | 0.87 |
| | November | -0.24 | -0.27 | -0.21 | 0.78 |
| | December | -0.65 | -0.67 | -0.62 | **0.52** |

Values in bold indicate lowest and highest odds ratio in each category.

[57]. Clearly, using a model that can accurately predict attendance probabilities in the design of such interventions would increase their cost-effectiveness.

Fig 2 shows the outcome when patients are assigned, in increasing order of attendance probability (calculated in three different ways: by LASSO, by Random Forest, and at random) to different sizes of intervention target group. For example, if it is only possible to include 30% of all patients in the intervention group, nevertheless over 70% of the no-show patients in our

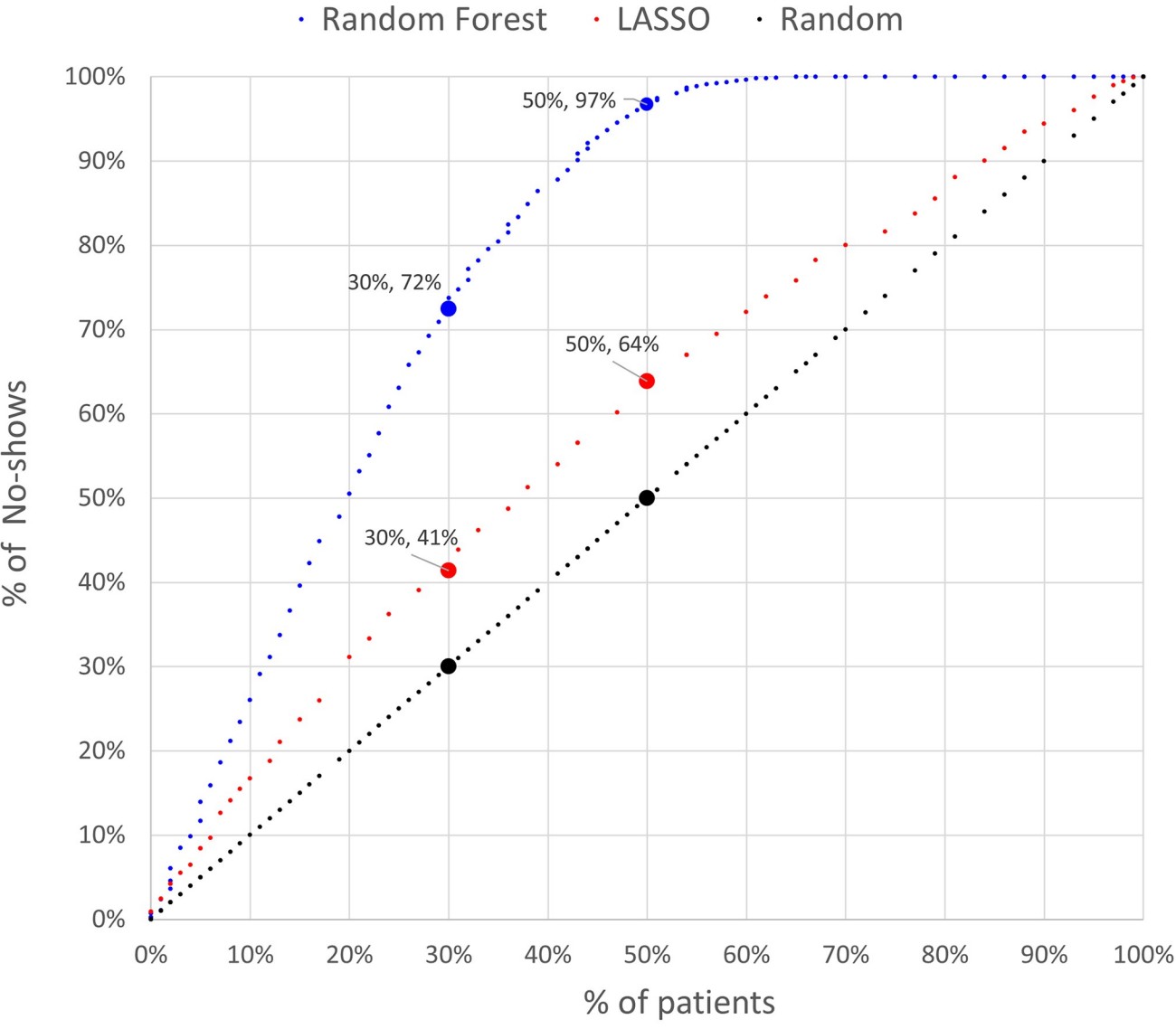

**Fig 2. Model performance.**

data would receive the intervention using the RF classification. This coverage would decrease to 41% if the LASSO model was used, and just 30% if the decision was made without the support of a classification model (i.e., patients were assigned to the intervention group at random). Conversely, we can also use Fig 2 to quantify the risk of a classification model, defined as the percentage of no-show patients who do not receive the intervention. For example, suppose the intervention is able to reach 50% of all patients. If the RF were used to make the selection, only 3% (100% - 97%) of the no-show patients would not have been included. This percentage would increase to 36% using LASSO, or 50% if patients were classified at random.

### 3.2. Qualitative results

The aim of the interviews was to understand attendance barriers among patients with high and medium risk of no-show, as well as to identify some perceived benefits of the cytology and

**Table 5. Quotes from the interviews.**

| N | Category | Quote | Frequency |
|---|----------|-------|-----------|
| 2 | Category: Barriers—Access Inconvenient appointment slots | *I would say that [it is important to have] more service time. Sometimes you go to work at five or four thirty in the morning and you are back home at seven p.m. There is not service at nights and weekend appointments are always booked. It is difficult to keep an appointment* | 5 |
| 3 | Category: Barriers—Access Long lead times | *If you go [to the healthcare facility] they say that you need to call [to book an appointment]. Then you call, and they say there are not available slots. After a time, you just get tired and stop trying.* | 12 |
| 15 | Category: Barriers—Service delivery Process design | *It is not always clear what you need to do. Sometimes you need to carry out administrative paperwork and spend almost all day waiting in queues* | 18 |
| | | *You need to go through administrative clearance for almost everything! I even took a mammogram a while ago and have no idea how to get the results or book an appointment.* | |
| 16 | Category: Barriers—Personal Family care | *I have three children. For their appointments, I normally ask for some time off work. If I do the same [for mine] they would say I am always out. That is problematic.* | 11 |
| | | *If you are a mom with small children, sometimes you just do not find anyone to take care of them* | |
| 17 | Category: Barriers—Personal Forgetfulness | *Sometimes you forget because you are caught in the middle of so many things to do. It would be good if someone calls you to remind the appointment.* | 11 |
| 28 | Category: Barriers—Protective behaviour Embarrassment | *As a woman, I am embarrassed that someone examines that part of my body* | 5 |
| 29 | Category: Barriers—Protective behaviour Gender of the health provider | *Once I saw that a male nurse was performing the cytology at that facility. I decided to miss my appointment. I prefer to be examined by a woman* | 2 |
| 32 | Category: Benefits—Protective behaviour Cancer diagnosis | *It seems to me that having a cytology is essential. It is a way of preventing cancer and knowing what diseases one might have.* | 17 |
| 37 | Category: Benefits—Service delivery Satisfaction (home visit) | *The visit went well. She [the community worker] was kind, took my blood pressure and my weight. She even helped me with some appointments I needed* | 28 |
| 38 | Category: Benefits—Service delivery Satisfaction (service delivery) | *So far, the doctors I have seen are really good. I have been operated, hospitalized and the service is always good. I have felt supported* | 32 |
| 39 | Category: Susceptibility Perceived susceptibility | *It is important to have a cytology because one might develop cancer. My daughter was infected with human papillomavirus a while ago. She was timely diagnosed and thanks to God, there were no other consequences.* | 4 |
| 41 | Category: Severity Fear of a bad result | *Sometimes women are scared about getting a bad result.* | 7 |
| 42 | Category: Severity Fear of side effects | *I have heard that some healthy women end up with infections and bleeding after the cytology.* | 5 |

outreach programs. First-order categories are illustrated by quotes extracted from the interviews in Table 5. The final column of Table 5 shows the number of interviews in which each category was coded. In the rest of this section, we present the main qualitative findings.

Participants found it hard to navigate the service delivery process (see code 15 in Table 5). They felt that when attending a medical appointment, most of the time was spent in the waiting room or carrying out administrative paperwork. They also reported that it was common to have to provide the same information more than once to different staff within the same healthcare facility, or even to miss appointments because they were not properly briefed about the necessary administrative or clinical requirements. For example, some participants reported that even though they attended, they were not examined because they had had sexual intercourse the previous night. Lastly, a small number of participants commented on the (perceived) low quality of care they had experienced using that healthcare service.

There were barriers to accessing healthcare services. The most commonly raised concern was that it was difficult to book an appointment because lead times were long, healthcare facilities had inconvenient opening hours and call centres were permanently busy. This is particularly relevant in a context where most patients have informal jobs and are unable to attend appointments in working hours. For this reason, other participants mentioned difficulties in taking time off work, financial pressures, and problems with transport. Some quotes from interviewees affected by such problems are presented under categories 2 and 3 of Table 5.

Personal problems and beliefs about cervical cytology could also lead to a missed appointment (see quotes under categories 18, 19, 28 and 29 in Table 5). The most common personal problems were forgetfulness and family care responsibilities. Among the latter, some participants reported that they tend to prioritize medical appointments for other members of their family or were not always able to find someone to take care of their children during the appointments. Regarding the cytology test itself, some participants believed that the procedure would be painful or uncomfortable, or that they would feel anxiety or embarrassment. Moreover, some said that they were not able to attend because they were menstruating or had had sexual intercourse the day before the appointment. Two participants said that they decided not to attend because of the risk that a male nurse might examine them.

Despite the barriers described above, many participants reported that they were satisfied with the service they received, both in the healthcare facility and during the home visit (see categories 37 and 38 in Table 5). Most of them said that the community workers were kind and provided a direct way to overcome access barriers. Additionally, the home visits were informative. Most patients were aware of the purpose of the cytology test to diagnose cervical cancer and had a basic understanding of the screening program. However, some patients only had a general understanding of how screening could benefit their health, with no specific knowledge of the actual diseases that could be prevented.

Lastly, some comments related to susceptibility and severity. Some participants recognized that the purpose of the cytology test was to diagnose cancer, which could be interpreted as a sign of perceived susceptibility. However only three participants explicitly talked about their own risk of developing cancer. Moreover, those three participants had a family history of cancer or human papillomavirus infection. Additionally, fears of testing positive or of unpleasant side effects were stressed as possible reasons for missed cytology appointments. Table 5 presents some related quotes under categories 33, 40, 42 and 43.

## 4. Discussion

In this section we present a summary of our main findings, compare our study with others in the literature and consider implications for practice.

### 4.1. Main findings

Using routinely collected data, we were able to accurately predict individual attendance probabilities for cervical cancer screening appointments in Bogotá. First, we fitted a LASSO regression model to identify the characteristics of the higher no-show risk appointments. We found that younger patients living in zones with higher poverty levels are less likely to attend. Additionally, offering short lead times and Sunday appointments could increase screening uptake among hard-to-reach women in the city. Next, we used the LASSO results to select the variables to train an RF aimed at improving prediction accuracy. The resulting model has a good discrimination power and low variability in its performance (Average AUROC score 0.84 and standard deviation of 0.01). We used the RF results to inform the sample selection for a series of semi-structured interviews.

We interviewed 60 hard-to-reach women who received a home visit from the outreach program and had failed to attend their cytology appointments. Although most patients perceived the home visits to be informative, they found it hard to navigate the service delivery process and experienced access barriers. Qualitative data also enhanced the interpretation of the quantitative results. For example, the LASSO results show a relationship between the appointment date and the attendance probability. In the same vein, during the interviews some patients

expressed that taking time off from work or childcare responsibilities might act as deterrents for screening uptake.

## 4.2. Comparison with other studies

Two of our quantitative results confirm what has been found in other cytology uptake studies: attendance probabilities change with the patient age and poverty. In Bogotá, we find the younger the patient, the higher her no-show risk. While some studies report similar behaviour in Ethiopia [58], or Kenya [59], in Tanzania younger patients are more likely to keep their appointments [60]. Since this finding is context-dependent, it highlights the relevance of conducting research to inform public policy. We also find that, in zones where poverty affects less than 18% of the population, patients are three times more likely to attend. Several other studies have identified the same relationship between poverty and cervical cancer screening [59–62]. Moreover, during the interviews, some participants reported financial and transport difficulties in attending. Our qualitative results confirm the quantitative findings regarding financial difficulties.

We found a statistical relationship between the appointment date, i.e., day of week and month of year, and the attendance probability. In cervical cancer screening, most previous research has been devoted to exploring the impact of socio-demographic variables on attendance for screening [10]. The databases analysed in such studies normally include patients that do not have scheduled appointments. As our research was conducted within an outreach program, the context is slightly different. However, patients' lack of time has been described as a barrier for cytology uptake [63, 64]. Our results can be also compared with previous work in no-show risk for primary care appointments. The existence of patterns in attendance probabilities according to the month of the year or day of the week has been documented previously [65–68]. Two qualitative results might offer a context for these quantitative findings. First, a lower no-show risk on Sundays might be explained by the difficulties reported by some participants in taking time off work. Second, some participants stated that their childcare responsibilities caused them to miss appointments, which could explain why the no-show risk was higher during the school vacation months.

In our quantitative analysis, the attendance probability increases with the lead time. This is also found to be the case in studies of no-show behaviour for other primary care appointments [24, 69]. Even in contexts where cultural barriers towards cervical cancer screening are overcome using education campaigns, offering timely access is a key component to increasing coverage [2, 70]. Confirming this quantitative finding, many of our participants stated that booking appointments is hard. They said that lead times were long (which could increase forgetfulness), and the healthcare facilities had inconvenient opening hours. These access problems were also found as a relevant barrier for screening programs in five other Latin American countries [71].

Our qualitative study showed that participants found the home visits to be informative. Unsurprisingly, therefore, most participants either said they were aware that cytology is used to diagnose cancer or recognised the importance of regular cervical cytology. This result, however, is different from what has been found in many developing countries. Lack of knowledge regarding cervical cancer and screening programs has been identified as a key predictor of low uptake rates (see Table 3). Nevertheless, it must be noted that some participants explicitly stated that the importance of cytology was not discussed during the home visit. Additionally, only a few participants considered themselves personally to be at risk of developing cervical cancer. The belief that a disease is something faced by other people and not oneself is described by [72] as *"othering"*, and leads people to underestimate the prevalence of the disease.

### 4.3. Implications for program management and public policy

Our findings suggest a lack of coordination between the two components of the screening program, home visits and screening appointments. A great effort is made by the home visits team to reach patients who need screening, but although these patients are willing to take part in the screening program, they still face several access barriers. There is a need to offer agile scheduling and cancellation systems, shorter lead times and more flexible opening hours. Therefore, capacity management practices should be reviewed. Alternatives to the current operation might include: performing the cytology during the home visit [73], minimizing the impact of no-shows by overbooking [65], pooling resources within the program [74] and offering open access scheduling policies for some healthcare facilities [75].

There is potential to overcome most of the perceived barriers by improving the service delivery process. On the one hand, community workers have a unique knowledge and understanding of the cultural context of the patient [76]. Home visits could provide more standardized information about cervical cancer, the screening program, and the best way to navigate the healthcare system. Therefore, educational interventions for community workers [77] and better design of information material for patients [78–80] could provide an interesting opportunity for the outreach program. On the other hand, drawing from our participants' experiences, there is a clear need to improve service quality at the healthcare facilities. Mechanisms to reduce waiting times for cytology [81, 82] and to improve the organization of the program [83, 84] could increase satisfaction and attendance levels.

An impact evaluation should support future decision-making. After three years of running the outreach program, the District Secretariat of Health (*Secretaría Distrital de Salud*, SDS) has sufficient information to quantify its achievements in terms of early diagnosis. However, a model-based evaluation would enable different policy alternatives to be compared [85, 86]. While in developed countries cytology-based programs have achieved good results in decreasing morbidity and mortality of cervical cancer, in developing countries this is not always the case [76]. A combination of alternative cervical cancer screening tests could enhance capacity and improve health outcomes in low resourced health systems [87]. For example, a recent review concluded that self-sampling approaches have been found to increase acceptability of cervical cancer screening [88]. In this context, by modelling the patient pathway from the home visit to treatment completion, a simulation model could support resource allocation and inform policy design.

### 4.4. Limitations

At the time of writing, three main limitations of this study are being addressed in ongoing projects. First, since we are using only routinely-collected quantitative data, the quantitative models predict attendance probabilities based only on a set of variables that has been designed for administrative purposes. Therefore, it was not possible to quantify the relationship between attendance probabilities and other variables thought to be highly predictive, such as patient income or the time of day of the appointment [16, 89]. Second, for both phases, the sample is limited by the inclusion of women who have participated in the outreach program managed by SDS. Although this program covers most of the low-income women in the Bogotá, we do not know the perspectives or risk categories for other women in the city, or other parts of the country, that could inform public policy. Third, our findings suggest a relationship between the constructs of the HBM and no-show behaviour. However, this relationship is still to be quantified.

### 4.5. Conclusion

This study has shown the benefits of combining a 'black box' approach, machine learning, with an in-depth qualitative methodology that can explore, and potentially explain, the results

from the quantitative analysis. From a practical perspective, our findings indicate an urgent need to address the lack of alignment between the different phases of the cervical cancer screening program in Bogotá, and work to address this is currently under way.

## Supporting information

**S1 Appendix. Interview guide.**
(DOCX)

**S2 Appendix. First order categories description and references.**
(DOCX)

## Author Contributions

**Conceptualization:** David Barrera Ferro.

**Data curation:** Laura Bocanegra.

**Formal analysis:** David Barrera Ferro, Laura Bocanegra, Adriana Díaz, Elena Valentina Gutiérrez-Gutiérrez.

**Methodology:** David Barrera Ferro.

**Supervision:** Steffen Bayer, Sally Brailsford, Honora Smith.

**Writing – original draft:** David Barrera Ferro.

**Writing – review & editing:** Steffen Bayer, Sally Brailsford, Honora Smith.

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
