## [Decision Letter · Decision Letter 0]

13 Sep 2021

PONE-D-20-25493

Understanding no-show behaviour for cervical cancer screening appointments among low-income women in Bogotá, Colombia: a mixed-methods approach

PLOS ONE

Dear Dr. Barrera Ferro,

Thank you for submitting your manuscript to PLOS ONE. After careful consideration, we feel that it has merit but does not fully meet PLOS ONE’s publication criteria as it currently stands. Therefore, we invite you to submit a revised version of the manuscript that addresses the points raised during the review process.

We look forward to receiving your revised manuscript.

Kind regards,

Joel Msafiri Francis, MD, MS, PhD

Academic Editor

PLOS ONE

Journal Requirements:

2. Thank you for including your ethics statement: 'SDS (2019EE47807), Pontificia Universidad Javeriana (FID-19-107) and the University of Southampton (ERGO ID 48583.A1) granted ethical approval for this study.'   

(a) Please amend your current ethics statement to include the full name of the ethics committee/institutional review board(s) that approved your specific study.  

(b) Once you have amended this/these statement(s) in the Methods section of the manuscript, please add the same text to the “Ethics Statement” field of the submission form (via “Edit Submission”). 

4. In the ethics statement in the manuscript and in the online submission form, please provide additional information about the patient records used in your retrospective study, including: a) whether all data were fully anonymized before you accessed them; b) the date range (month and year) during which patients' medical records were accessed; c) the date range (month and year) during which patients whose medical records were selected for this study sought treatment; and d) the source of the medical records analyzed in this work (e.g. hospital, institution or medical center name).

5. In your Methods section, please provide additional information about the participant recruitment method and the demographic details of your participants. Please ensure you have provided sufficient details to replicate the analyses such as: a) the recruitment date range (month and year), b) a table of relevant demographic details, c) a statement as to whether your sample can be considered representative of a larger population, d) a description of how participants were recruited, and e) descriptions of where participants were recruited and where the research took place.

6. To comply with PLOS ONE submission guidelines, in your Methods section, please provide additional information regarding your statistical analyses. For more information on PLOS ONE's expectations for statistical reporting, please see https://journals.plos.org/plosone/s/submission-guidelines.#loc-statistical-reporting.

7. We note that the grant information you provided in the ‘Funding Information’ and ‘Financial Disclosure’ sections do not match. 

8. Thank you for stating in your Funding Statement: 

"The first author’s research is partially funded by a PhD scholarship from the healthcare research stream of the program Colombia Científica – Pasaporte a la Ciencia, granted by the Colombian Institute for Educational Technical Studies Abroad (Instituto Colombiano de Crédito Educativo y Estudios Técnicos en el Exterior, ICETEX)."

9. Please upload a new copy of Figure 4 as the detail is not clear. Please follow the link for more information: https://blogs.plos.org/plos/2019/06/looking-good-tips-for-creating-your-plos-figures-graphics/" https://blogs.plos.org/plos/2019/06/looking-good-tips-for-creating-your-plos-figures-graphics/.

10. We note that Figure 2 in your submission contain map images which may be copyrighted. All PLOS content is published under the Creative Commons Attribution License (CC BY 4.0), which means that the manuscript, images, and Supporting Information files will be freely available online, and any third party is permitted to access, download, copy, distribute, and use these materials in any way, even commercially, with proper attribution. For these reasons, we cannot publish previously copyrighted maps or satellite images created using proprietary data, such as Google software (Google Maps, Street View, and Earth). For more information, see our copyright guidelines: http://journals.plos.org/plosone/s/licenses-and-copyright.

Natural Earth (public domain): http://www.naturalearthdata.com/.

Reviewers' comments:

Reviewer's Responses to Questions

**Comments to the Author**

1. Is the manuscript technically sound, and do the data support the conclusions?

Reviewer #1: Partly

Reviewer #2: Yes

2. Has the statistical analysis been performed appropriately and rigorously? 

Reviewer #1: No

Reviewer #2: I Don't Know

3. Have the authors made all data underlying the findings in their manuscript fully available?

Reviewer #1: No

Reviewer #2: Yes

4. Is the manuscript presented in an intelligible fashion and written in standard English?

Reviewer #1: Yes

Reviewer #2: Yes

5. Review Comments to the Author

Reviewer #1: Thank you for the opportunity review this article. It is important and its findings merit publication. This reviewer has a few concerns regarding the following broad areas: (1) there are few details regarding the screening program eligibility criteria and these details are needed for interpretation of your results; (2) the prediction model is interesting, however, given the limited variables used as inputs – the paper should de-emphasize the results of the model and underscore its use as way to identify women for the qualitative interviews; (3) authors should focus more on the qualitative results and improve the interpretability of Figure 4 – currently, this reviewer does not understand its utility; and (4) the literature review is useful, however, please expand to include PubMed as I believe the authors are missing quite a few relevant pubs.

Specific comments:

• The introduction includes the necessary points and rationale for conducting the study. However, it could be better organized. Please ensure that the final paragraph of the introduction outlines the objectives of the study. Lines 68-87 could be moved to the discussion.

• Methods

o Lines 110-111: Details regarding the program would be useful here such as: eligibility criteria for women, what areas of Colombia are included in this program (i.e. was it just Bogota or other cities in Colombia as well), how many women were eligible based on census data, what percentage of women were contacted for appointments, how were the woman contacted (door to door or via telephone, or both)? Regarding eligibility criteria for women – can the authors also comment on how they define low-income per the programs description?

o Line 110: of the 23,384 appointments scheduled, were these all unique patient appointments or might have there been repeats? If there are repeats, is it possible to get the number of unique appointments?

o Lines 128 – 131: Can you provide the number of women that fell into each category of the eligibility criteria so the reader can get a sense of each criterion?

o Table 2: In Line 131, the text mentions 75 participants were included. However, Table 2 mentions 60 interviews. Please clarify this discrepancy.

o Table 3: Please provide a footnote that defines “Inductive category” clearly. From the text, I believe it means an emerging theme or an understudies topic.

o Line 153: I am concerned that the SCOPUS database only retrieved 394 records when the authors searched cervical cancer screening and uptake. I did the same search in PubMed and found over 1500 results. Can the authors include PubMed given the comprehensive nature of this database? I find it difficult to believe that there were no papers that identified “cancer diagnosis” or “health” as a potential benefit of cancer screening.

o Table 2: Can the authors describe what “conditions” means under protective behavior?

o Line 165 – typo

• Results

o Lines 208-213: A comment here regarding the validity of the model is needed. The patient specific inputs are very few. More data regarding other factors such as marital status, number of children, employment status, educational level etc. are needed for this prediction model to be useful.

o Table 4: Is it possible to create more categories of Poverty? Why did the authors decide to create only a binary variable? It would be important to show that high poverty areas were less likely to attend.

o Line 215: What is “RF”?

o Figure 3 is not useful. I am not sure how to interpret it given how unreliable the model is. I would suggest the authors remove this and delete text 214 – 223.

o Figure 4 is very blurry. Can the authors provide some footnotes on Figure 4 to explain why it is arranged the way it is? Do the boxes mean something? Is there a reason some boxes are shaped differently than others? The figure is not self explanatory and should be able to stand alone.

• Discussion

o Line 271; Summary of overall main findings would be useful here.

o Line 329: Define SDS

Reviewer #2: Thank you for the opportunity to review this manuscript. This is a well written manuscript with an interesting research question and methodological approach. It is great to see mixed-methods approach being used in this setting. The authors have done a nice job in describing the methods and results. Non-compliance to cervical cancer screening is furthermore an important topic to address and understand, in order to design future policies to improve compliance rates.

Minor questions that may be adressed:

1) For the qualitative interviews, what questions were asked? How did you design the interview guide/questions?

2) In Table 3, could you add more description to the first order categories? These are interesting and may be useful to others; however, while some are self explanatory, others are not.

3) Results Table 4, could you elaborate what drives the differences between days and months?

4) The discussion section would benefit from a brief summary of main findings at the beginning, including a discussion of how the qualitative and quantitative methods complemented each other.

5) In the discussion of implications and future policies, could HPV self-sampling be an option? Home-based self-collection of samples is currently being rolled out worldwide as an approach to reach underscreened women. Is this relevant for this setting? What is the status of HPV testing?

6. PLOS authors have the option to publish the peer review history of their article (what does this mean?). If published, this will include your full peer review and any attached files.

Reviewer #1: **Yes: **Jessica Islam

Reviewer #2: No

---

## [Author Response · Author response to Decision Letter 0]

7 Oct 2021

Editor 

Comment:

Thank you for submitting your manuscript to PLOS ONE. After careful consideration, we feel that it has merit but does not fully meet PLOS ONE’s publication criteria as it currently stands. Therefore, we invite you to submit a revised version of the manuscript that addresses the points raised during the review process

Reply:

Thank you for your encouragement and the opportunity to review our work. We have made extensive revisions to the document which are detailed in this letter, which we are hopeful will satisfy the reviewers’ concerns.

Journal Requirements

Comment 1:

Reply 1:

We have reviewed our manuscript to comply with format requirements.

Comment 2:

Thank you for including your ethics statement: 'SDS (2019EE47807), Pontificia Universidad Javeriana (FID-19-107) and the University of Southampton (ERGO ID 48583.A1) granted ethical approval for this study.' 

(a) Please amend your current ethics statement to include the full name of the ethics committee/institutional review board(s) that approved your specific study. 

(b) Once you have amended this/these statement(s) in the Methods section of the manuscript, please add the same text to the “Ethics Statement” field of the submission form (via “Edit Submission”). 

Reply 2:

We have included additional details on our methods section. Lines 92-95.

Comment 3:

Please provide additional details regarding participant consent. In the ethics statement in the Methods and online submission information, please ensure that you have specified (1) whether consent was informed and (2) what type you obtained (for instance, written or verbal, and if verbal, how it was documented and witnessed). If your study included minors, state whether you obtained consent from parents or guardians. If the need for consent was waived by the ethics committee, please include this information.

Reply 3:

All participants granted informed consent during the phone call, and it was recorded. We have added a clarification on Section 2.3. Lines 174-176

Comment 4:

In the ethics statement in the manuscript and in the online submission form, please provide additional information about the patient records used in your retrospective study, including: a) whether all data were fully anonymized before you accessed them; b) the date range (month and year) during which patients' medical records were accessed; c) the date range (month and year) during which patients whose medical records were selected for this study sought treatment; and d) the source of the medical records analyzed in this work (e.g. hospital, institution or medical center name).

Reply 4:

We have added additional details in Section 2.2. Lines 134-144

Comment 5:

In your Methods section, please provide additional information about the participant recruitment method and the demographic details of your participants. Please ensure you have provided sufficient details to replicate the analyses such as: a) the recruitment date range (month and year), b) a table of relevant demographic details, c) a statement as to whether your sample can be considered representative of a larger population, d) a description of how participants were recruited, and e) descriptions of where participants were recruited and where the research took place.

Reply 5:

We have added additional details in Section 2.3. Lines 163-170

Comment 6:

To comply with PLOS ONE submission guidelines, in your Methods section, please provide additional information regarding your statistical analyses. For more information on PLOS ONE's expectations for statistical reporting, please see https://journals.plos.org/plosone/s/submission-guidelines.#loc-statistical-reporting.

Reply 6:

We have used RECORD (The REporting of studies Conducted using Observational Routinely-collected health Data) guidelines. We have also provided some additional information to clarify that our methods are reproducible and included the reference for the Python implementation. Lines 146-158.

Comment 7:

We note that the grant information you provided in the ‘Funding Information’ and ‘Financial Disclosure’ sections do not match. 

Reply 7:

Thank you for the opportunity to clarify. The funding statement was included in the cover letter. 

Comment 8:

Thank you for stating in your Funding Statement: 

"The first author’s research is partially funded by a PhD scholarship from the healthcare research stream of the program Colombia Científica – Pasaporte a la Ciencia, granted by the Colombian Institute for Educational Technical Studies Abroad (Instituto Colombiano de Crédito Educativo y Estudios Técnicos en el Exterior, ICETEX)."

Reply 8:

Thank you for the opportunity to clarify. The funding statement was included in the cover letter. 

Comment 9:

Please upload a new copy of Figure 4 as the detail is not clear.

Reply 9:

Upon consideration, we have decided to eliminate Figure 4.

Comment 10:

We note that Figure 2 in your submission contain map images which may be copyrighted. All PLOS content is published under the Creative Commons Attribution License (CC BY 4.0), which means that the manuscript, images, and Supporting Information files will be freely available online, and any third party is permitted to access, download, copy, distribute, and use these materials in any way, even commercially, with proper attribution. For these reasons, we cannot publish previously copyrighted maps or satellite images created using proprietary data, such as Google software (Google Maps, Street View, and Earth). For more information, see our copyright guidelines: http://journals.plos.org/plosone/s/licenses-and-copyright.

Reply 10:

Upon consideration, we have decided to eliminate Figure 2.

Reviewer #1: 

Comment 1:

Thank you for the opportunity review this article. It is important and its findings merit publication. 

Reply 1:

Thank you for review and detailed feedback. We have addressed all your comments as explained below.

Comment 2:

This reviewer has a few concerns regarding the following broad areas:

(1) there are few details regarding the screening program eligibility criteria and these details are needed for interpretation of your results.

(2) the prediction model is interesting, however, given the limited variables used as inputs – the paper should de-emphasize the results of the model and underscore its use as way to identify women for the qualitative interviews.

(3) authors should focus more on the qualitative results and improve the interpretability of Figure 4 – currently, this reviewer does not understand its utility. 

(4) the literature review is useful, however, please expand to include PubMed as I believe the authors are missing quite a few relevant pubs.

Reply 2:

Thank you for this summary. It was very helpful to understand the main concerns about our manuscript. We address each of these four points on the detailed comments below. Broadly, we have made the following changes:

(1) We included details of both the nation-wide screening program and the outreach program.

(2) We improved the description of the prediction algorithm. 

(3) Upon consideration, we decided to eliminate Figure 4.

(4) We updated the review. 

Comment 3:

The introduction includes the necessary points and rationale for conducting the study. However, it could be better organized. Please ensure that the final paragraph of the introduction outlines the objectives of the study. Lines 68-87 could be moved to the discussion.

Reply 3:

We agree that closing the Introduction with the aims of the study improves the transition to the Methods section. This paragraph is now on lines 80-86. However, to comply with journal’s requirement of providing relevant literature within the Introduction, we have decided to keep lines 68-87 in this section.

Comment 4: Methods

Lines 110-111: Details regarding the program would be useful here such as: eligibility criteria for women, what areas of Colombia are included in this program (i.e. was it just Bogota or other cities in Colombia as well), how many women were eligible based on census data, what percentage of women were contacted for appointments, how were the woman contacted (door to door or via telephone, or both)? Regarding eligibility criteria for women – can the authors also comment on how they define low-income per the programs description?

Reply 4:

We have added a new Section 2.1. Lines 97-118

Comment 5: Methods

Line 110: of the 23,384 appointments scheduled, were these all unique patient appointments or might have there been repeats? If there are repeats, is it possible to get the number of unique appointments?

Reply 5:

We have added a clarification on Section 2.2. Lines 134-136

Comment 6: Methods

Lines 128 – 131: Can you provide the number of women that fell into each category of the eligibility criteria so the reader can get a sense of each criterion?

Reply 6:

We have added details Section 2.3. Lines 163-168

Comment 7: Methods

Table 2: In Line 131, the text mentions 75 participants were included. However, Table 2 mentions 60 interviews. Please clarify this discrepancy.

Reply 7:

Although 75 were reached by phone, only 60 accepted our invitation to take part of the study. We have highlighted relevant information on section 2.2. Lines 167-169.

Comment 8: Methods

Table 3: Please provide a footnote that defines “Inductive category” clearly. From the text, I believe it means an emerging theme or an understudies topic. 

Reply 8:

We agree this was not clear. We have added a footnote at the end of table 3. Line 222

Comment 9: Methods

Line 153: I am concerned that the SCOPUS database only retrieved 394 records when the authors searched cervical cancer screening and uptake. I did the same search in PubMed and found over 1500 results. Can the authors include PubMed given the comprehensive nature of this database? I find it difficult to believe that there were no papers that identified “cancer diagnosis” or “health” as a potential benefit of cancer screening.

Reply 9:

Many thanks for your comment and the opportunity to clarify. The main difference between the two results is that we searched using title and abstract of each paper. This change decreases the number of results from over 1500 to 535 using PubMed. Most of these 535 papers were either included in the SCOPUS data base or published after our submission. However, we updated the two reviews and made the following changes in the manuscript:

a. Added a clarification on Section 2.3. Lines 193-195.

b. Updated Figure 1. 

c. Updated the statistics on Section 2.3. Lines 196-199.

Regarding the two inductive categories, many researchers have identified the lack of perceived benefits or the lack of knowledge as possible deterrents for screening uptake. This could be related to the “Health” or “Cancer diagnosis” categories. However, our participants talked specifically about these concepts during the interviews. This could be related to the fact that all these patients have already received the home visit to talk about the outreach program. Therefore, we decided to add independent first order categories in the framework. 

Comment 10: Methods

Table 2: Can the authors describe what “conditions” means under protective behavior?

Reply 10:

We agree this particular term was unclear. We have changed first order category number 26 to “Non-compliance with requirements”

Comment 11: Methods

Line 165 – typo

Reply 11:

We have made changes on line 207

Comment 12: Results

Lines 208-213: A comment here regarding the validity of the model is needed. The patient specific inputs are very few. More data regarding other factors such as marital status, number of children, employment status, educational level etc. are needed for this prediction model to be useful.

Reply 12:

Many thanks for your comment and the opportunity to clarify. We aimed at using routinely collected data to predict individual no-show probabilities. To ensure that this model can be used by SDS, it was important to limit the input variables to those that are available in their information system. However, the discrimination power and the stability of the results lead us to conclude that this model can be used to generate highly accurate predictions and support resource-allocation decisions. 

In this context, we recognise that additional variables can be useful to improve the accuracy of the LASSO model. Additionally, we agree that our manuscript was lacking some details regarding the model to ensure that the reader can trust the results. Therefore, we have made the following changes:

a. We included a sentence recognizing the limitation to include other variables on Section 2.2. Lines 137-138.

b. We improved the description of our quantitative methods on Section 2.2. Lines 146-158.

c. We improved our results including comments on the performance of the models. Section 3.1, lines 229-239

d. We included information about the variability of the coefficients on Table 4

Comment 13: Results

Table 4: Is it possible to create more categories of Poverty? Why did the authors decide to create only a binary variable? It would be important to show that high poverty areas were less likely to attend.

Reply 13:

We used decision trees to automatically build the categories (the number and the limits) for each variable. We have added details on Section 2.2. Lines 139-142.

Comment 14: Results

Line 215: What is “RF”?

Reply 14:

We have ensured that the abbreviation was defined the first time we used it in section 2.3 (line 147) and included the full name in section 3.1 (line 261).

Comment 15: Results

Figure 3 is not useful. I am not sure how to interpret it given how unreliable the model is. I would suggest the authors remove this and delete text 214 – 223.

Reply 15: 

We hope that the changes we have made to improve the description of our modelling approach address the question of the reliability of our results. We have also made changes on Figure 3 (now Figure 2).

Comment 16: Results

Figure 4 is very blurry. Can the authors provide some footnotes on Figure 4 to explain why it is arranged the way it is? Do the boxes mean something? Is there a reason some boxes are shaped differently than others? The figure is not self explanatory and should be able to stand alone.

Reply 16:

Upon consideration we decided to eliminate this figure. We have added a column in Table 5 presenting the frequency of each category in our data.

Comment 17: Discussion

Line 271; Summary of overall main findings would be useful here.

Reply 17:

We have added a new Section 4.1. Lines 320-335

Comment 18: Discussion

Line 329: Define SDS

Reply 18:

We have ensured that the abbreviation was defined the first time we used it in section 1 (line 53) and included the full name in section 4.3 (line 391).

Reviewer #2: 

Comment 1:

Thank you for the opportunity to review this manuscript. This is a well written manuscript with an interesting research question and methodological approach. It is great to see mixed-methods approach being used in this setting. The authors have done a nice job in describing the methods and results. Non-compliance to cervical cancer screening is furthermore an important topic to address and understand, in order to design future policies to improve compliance rates.

Reply 1:

Thank you for your encouragement and positive feedback. We have addressed all your comments in detail as explained below.

Comment 2:

For the qualitative interviews, what questions were asked? How did you design the interview guide/questions?

Reply 2:

We have added details in Section 2.4. Lines 169-171

Comment 3:

In Table 3, could you add more description to the first order categories? These are interesting and may be useful to others; however, while some are self explanatory, others are not.

Reply 3:

We agree this would help, and have decided to include a new Appendix with the definition of all first order categories. We refer to the appendix on line 208.

Comment 4:

Results Table 4, could you elaborate what drives the differences between days and months?

Reply 4:

We discussed this in Sections 3.1 (Lines 246-250 ) and 4.2 (Lines) 

Comment 5:

The discussion section would benefit from a brief summary of main findings at the beginning, including a discussion of how the qualitative and quantitative methods complemented each other.

Reply 5:

We have added a new Section 4.1. Lines 320-333.

Comment 6:

In the discussion of implications and future policies, could HPV self-sampling be an option? Home-based self-collection of samples is currently being rolled out worldwide as an approach to reach underscreened women. Is this relevant for this setting? What is the status of HPV testing?

Reply 6:

Many thanks for your observation. We have made the following two changes.

a) We added a paragraph on Section 2.1, describing the status of the screening program including new legislation on HPV testing. Lines 97-108.

b) We made changes on Section 4.2, including a recent review on self-sampling for HPV testing. Lines 398-400

---

## [Decision Letter · Decision Letter 1]

8 Apr 2022

PONE-D-20-25493R1Understanding no-show behaviour for cervical cancer screening appointments among hard-to-reach women in Bogotá, Colombia: a mixed-methods approachPLOS ONE

Dear Dr. Barrera Ferro,

Thank you for submitting your manuscript to PLOS ONE. After careful consideration, we feel that it has merit but does not fully meet PLOS ONE’s publication criteria as it currently stands. Therefore, we invite you to submit a revised version of the manuscript that addresses the points raised during the review process.

We look forward to receiving your revised manuscript.

Kind regards,

Joel Msafiri Francis, MD, MS, PhD

Academic Editor

PLOS ONE

Journal Requirements:

Reviewers' comments:

Reviewer's Responses to Questions

**Comments to the Author**

1. If the authors have adequately addressed your comments raised in a previous round of review and you feel that this manuscript is now acceptable for publication, you may indicate that here to bypass the “Comments to the Author” section, enter your conflict of interest statement in the “Confidential to Editor” section, and submit your "Accept" recommendation.

Reviewer #3: All comments have been addressed

Reviewer #4: (No Response)

2. Is the manuscript technically sound, and do the data support the conclusions?

Reviewer #3: Yes

Reviewer #4: Yes

3. Has the statistical analysis been performed appropriately and rigorously? 

Reviewer #3: Yes

Reviewer #4: Yes

4. Have the authors made all data underlying the findings in their manuscript fully available?

Reviewer #3: Yes

Reviewer #4: No

5. Is the manuscript presented in an intelligible fashion and written in standard English?

Reviewer #3: Yes

Reviewer #4: Yes

6. Review Comments to the Author

Reviewer #3: Thank you for the opportunity to review the revised version of this manuscript. The authors’ responses to the comments from previous reviewers were satisfatory.

This exploratory sequential mixed methods study was meticulously conducted. The quantitative component interestingly applied machine learning techniques, the qualitative component was properly conducted, whereas the systematic review was adequately performed using two major literature databases, all of which were presented in compliance with standard reporting guidelines.

Nonetheless, I strongly believe that the references could be more concise—at least by moving several of them in the systematic review component to be supplementary file.

Reviewer #4: The authors have done an excellent job of responding to reviewer comments and the revisions have resulted in a presentation of the results that is clearer to the reader. There are a few minor points that I think would improve the structure and flow of the paper.

Lines 97-118 have been added in response to Reviewer 1; these are quite lengthy and can be summarized to focus on the clinical guidelines and current situation that may impact the study question and outcomes.

The results section (Starting in line 229) would be stronger if the authors started with a more clear description of their findings, rather than an explanation of the model. Lines 229-232 can all be placed in the methods.

Throughout the methods, there are opportunities to more clearly present what the findings show. Have the authors considered rephrasing the findings so that the odds ratios represent a greater odds of attending appointments? Rather than the counterintuitive “higher odds ratio=lower no show.” The other option is to flip the outcome variable, so that no-show is the outcome of interest, and the odds ratios would be flipped (i.e. a greater odds ratio would mean a greater odds of no-show).

In addition, the authors state that some restrictions apply to the data, likely the qualitative data.

7. PLOS authors have the option to publish the peer review history of their article (what does this mean?). If published, this will include your full peer review and any attached files.

Reviewer #3: **Yes: **Assoc. Prof. Dr. Krit Pongpirul, MD, MPH, PhD.

Reviewer #4: No

---

## [Author Response · Author response to Decision Letter 1]

17 May 2022

Editor

Comment:

Thank you for submitting your manuscript to PLOS ONE. After careful consideration, we feel that it has merit but does not fully meet PLOS ONE’s publication criteria as it currently stands. Therefore, we invite you to submit a revised version of the manuscript that addresses the points raised during the review process

Reply:

Thank you for your encouragement and the opportunity to review our work. We have made revisions to the document which are detailed in this letter. We are hopeful this version of the manuscript will satisfy the reviewers’ concerns.

Reviewer #3

Comment 1:

Thank you for the opportunity to review the revised version of this manuscript. The authors’ responses to the comments from previous reviewers were satisfactory. This exploratory sequential mixed methods study was meticulously conducted. The quantitative component interestingly applied machine learning techniques, the qualitative component was properly conducted, whereas the systematic review was adequately performed using two major literature databases, all of which were presented in compliance with standard reporting guidelines.

Reply 1:

Thank you for your assessment and positive feedback. We are glad to know that you found our responses satisfactory. We’ve made changes to address your concern.

Comment 2:

I strongly believe that the references could be more concise—at least by moving several of them in the systematic review component to be supplementary file.

Reply 2:

Upon consideration, we agree that it is possible to place the references supporting the first-order categories in a supplementary file, without making sacrifices on the overall quality of the paper. Hence, we have made two changes:

1. We changed Table 3 to present only the Conceptual Framework. Page 10.

2. We have added Table 2 to the Appendix S2

Reviewer #4

Comment 1:

The authors have done an excellent job of responding to reviewer comments and the revisions have resulted in a presentation of the results that is clearer to the reader. There are a few minor points that I think would improve the structure and flow of the paper.

Reply 1:

Thank you for assessment and the opportunity to review our work. In what follows, we address your concerns. 

Comment 2:

Lines 97-118 have been added in response to Reviewer 1; these are quite lengthy and can be summarized to focus on the clinical guidelines and current situation that may impact the study question and outcomes.

Reply 2:

Upon consideration we agree that some of the details could be deleted. We have rewritten this section to convey two main messages: i) the way the screening program works in Colombia and ii) the definition of hard-to-reach women adopted in Bogotá.

Comment 3:

The results section (Starting in line 229) would be stronger if the authors started with a more clear description of their findings, rather than an explanation of the model. Lines 229-232 can all be placed in the methods.

Reply 3:

We have placed lines 229-232 in the methods section. 

Comment 4: 

Throughout the methods, there are opportunities to more clearly present what the findings show. Have the authors considered rephrasing the findings so that the odds ratios represent a greater odds of attending appointments? Rather than the counterintuitive “higher odds ratio=lower no show.” The other option is to flip the outcome variable, so that no-show is the outcome of interest, and the odds ratios would be flipped (i.e. a greater odds ratio would mean a greater odds of no-show).

Reply 4:

You are right. To avoid any confusion, as the attendance was coded with 1, we have made changes through the paper stating that we aim at predicting attendance probabilities. This way, the aim and the variable representation are coherent.

---

## [Decision Letter · Decision Letter 2]

11 Jul 2022

Understanding no-show behaviour for cervical cancer screening appointments among hard-to-reach women in Bogotá, Colombia: a mixed-methods approach

PONE-D-20-25493R2

Dear Dr. Barrera Ferro,

We’re pleased to inform you that your manuscript has been judged scientifically suitable for publication and will be formally accepted for publication once it meets all outstanding technical requirements.

Kind regards,

Joel Msafiri Francis, MD, MS, PhD

Academic Editor

PLOS ONE

Additional Editor Comments (optional):

Reviewers' comments:

Reviewer's Responses to Questions

**Comments to the Author**

1. If the authors have adequately addressed your comments raised in a previous round of review and you feel that this manuscript is now acceptable for publication, you may indicate that here to bypass the “Comments to the Author” section, enter your conflict of interest statement in the “Confidential to Editor” section, and submit your "Accept" recommendation.

Reviewer #3: All comments have been addressed

2. Is the manuscript technically sound, and do the data support the conclusions?

Reviewer #3: Yes

3. Has the statistical analysis been performed appropriately and rigorously? 

Reviewer #3: Yes

4. Have the authors made all data underlying the findings in their manuscript fully available?

Reviewer #3: Yes

5. Is the manuscript presented in an intelligible fashion and written in standard English?

Reviewer #3: Yes

6. Review Comments to the Author

Reviewer #3: The responses are satisfactory. Thank you very much for revising the manuscript. I believe the current version is ready for publication.

7. PLOS authors have the option to publish the peer review history of their article (what does this mean?). If published, this will include your full peer review and any attached files.

Reviewer #3: **Yes: **Assoc. Prof. Dr. Krit Pongpirul, MD, MPH, PhD.

---

## [Editor Report · Acceptance letter]

14 Jul 2022

PONE-D-20-25493R2 

Understanding no-show behaviour for cervical cancer screening appointments among hard-to-reach women in Bogotá, Colombia: a mixed-methods approach 

Dear Dr. Barrera Ferro:

I'm pleased to inform you that your manuscript has been deemed suitable for publication in PLOS ONE. Congratulations! Your manuscript is now with our production department. 

Kind regards, 

on behalf of

Dr. Joel Msafiri Francis 

Academic Editor

PLOS ONE